# Functional and Antioxidant Properties of Plastic Bottle Caps Incorporated with BHA or BHT

**DOI:** 10.3390/ma14164545

**Published:** 2021-08-13

**Authors:** Yu-Wen Wang, Ya-Na Li, Qin-Bao Lin, Xiao Wang, Zeng-Hui Li, Kai-Xuan Wu

**Affiliations:** 1Department of Mechanical Engineering, Wuhan Polytechnic University, Wuhan 430023, China; wywwhpu@163.com (Y.-W.W.); wx774911157@163.com (X.W.); lzh1663186826@163.com (Z.-H.L.); wkx22527589421996@163.com (K.-X.W.); 2Packaging Engineering Institute, Jinan University, Zhuhai 519070, China

**Keywords:** plastic bottle caps, high-density polyethylene, antioxidant, migration, HPLC

## Abstract

In this study, we prepared new antioxidant active plastic bottle caps by incorporating butylated hydroxyanisole (BHA) or butylated hydroxytoluene (BHT) and 2% (*w*/*w*) white masterbatch in high-density polyethylene (HDPE). Fourier-transform infrared (FT-IR) spectrometry revealed that the antioxidants and HDPE were uniformly mixed with noncovalent bonding. In addition, the differential scanning calorimetry (DSC) test revealed that the change in melting point and initial extrapolation temperature of the antioxidant active caps was not significant. Sensory evaluation and removal torque tests validated the suitability of the antioxidant active plastic bottle caps in industrial application. The antioxidant activity increased with a greater concentration of BHA and BHT incorporated in both antioxidant active caps (*p* < 0.05) and with more impact on the BHA cap compared to BHT cap in terms of antioxidant activity. Migration experiments for 10 days at 40 °C and 2 h at 70 °C showed that active antioxidants in the plastic bottle cap were more easily released into fatty foods and milk products that are highly sensitive to oxidation, and the migration of BHA and BHT did not exceed the maximum amount specified in (EC) No 1333/2008 (<200 mg/kg). As such, the antioxidant active plastic bottle caps inhibited oxidation, thereby ensuring higher food quality.

## 1. Introduction

Improvements in living standards and conscious eating have revolutionized the food industry. For instance, polyethylene terephthalate (PET) bottles have been popular in beverage packaging [1]. However, there is a sustained pressure to develop more functional and safer caps to replace the low-quality, ordinary plastic bottle caps [2,3,4]. Therefore, there has been a massive investment in the major domestic beverage packaging material, with the aim of developing diverse bottle caps that meet global standards. Therefore, we developed novel antioxidant plastic bottle caps and analyzed their suitability in beverage packaging. The modern consumer has a high regard for food safety [5,6]. Quality food should be nutritious and safe [7,8,9]. Food spoilage increases the cost of production and wastes food that would have otherwise been consumed. Antioxidants are additives that delay oxidative changes in food. They can prevent changes in taste and slow down rancidity, as well as discoloration, in food [10]. Compared to PET bottles, it is more practical to analyze the relatively flexible antioxidant active plastic bottle caps.

The major challenge associated with adopting food antioxidant packaging materials is an effective and practical combination of the antioxidant and packaging material without changing the packaging line or the characteristics of the materials [11,12]. Moreover, the concentration of antioxidant compounds should be optimal to maintain food quality and keep it fresh.

The most common synthetic phenolic antioxidants used in food preservation include butylated hydroxyanisole (BHA) and butylated hydroxytoluene (BHT). These antioxidants bind free radicals, remove oxygen, inhibit oxidases, and block oil and fat oxidation [13,14]. BHA and BHT residues in edible oils, fried foods, dried fish products, biscuits, instant noodles, canned fruits, and pickled meat products should not exceed 200 mg/kg. In food, they are often used alone or in combination with other compounds.

This study was aimed at developing antioxidant active plastic bottle caps using BHA and BHT. The effects of different BHA and BHT concentrations on the functional and antioxidant properties of caps were also tested.

## 2. Materials and Methods

### 2.1. Chemicals and Reagents

HDPE (HDPE 2200JP; Bangkok, Thailand) and masterbatch (R41411A; Dongguan, China) were obtained from Jinfu Technology Co., Ltd. (Dongguan, China). BHA (≥99%), while BHT (≥99%) was purchased from Shanghai Macklin Co., Ltd. (Shanghai, China). High-performance liquid chromatography (HPLC)-grade acetic acid, ethyl acetate ethanol, methanol, 2-diphenyl-1-picrylhydrazyl (DPPH) (96% free radical), vitamin C (Vc) (purity ≥ 99%), 2,4,6-Tris(2-pyridyl)-*s*-triazine (TPTZ), (purity ≥ 97%), and 2,2′-azinobis-(3-ethylbenzthiazoline-6-sulfonate) (ABTS) (purity ≥ 98%) were purchased from Macklin (Shanghai, China). Analytically pure (AR) potassium persulfate (K_2_S_2_O_8_) was purchased from Xilong Scientific Co., Ltd. (Guangdong, China), while AR sodium acetate trihydrate (C_2_H_3_NaO_2_·3(H_2_O)) was purchased from Guangdong Guangtry reagent Technology Co., LTD, (Guangdong, China). AR Ferric chloride (FeCl_3_) (99% pure) was purchased from Shanghai Macklin Co., Ltd. AR hydrochloric acid (HCl) was purchased from Guangdong Guangtry reagent Technology Co., LTD.

### 2.2. Sample Preparation

Synthetic antioxidants (BHA 1%, BHA 2%, BHT 1%, and BHT 2%) (*w*/*w*) were separately mixed with HDPE granules and 2% (*w*/*w*) masterbatch to obtain white granules. The mixture was transferred to a twin-screw extruder (MEDU-22/40, Putong, Guangzhou, China) using a screw diameter of 20 mm and 100 rpm screw speed. Temperatures in the nine heating zones of the rotating twin-screw extruder were 160 °C, 165 °C, 165 °C, 170 °C, 175 °C, 175 °C, 175 °C, and 175 °C. Molten extruded material was ejected as thin strings, passed through a cold-water basin, and then cut into small granules. The screw was set to rotate at 20 rpm, under temperatures of 190 °C, 200 °C, 210 °C, and 230 °C. The remaining granules were added into an injection molding machine (HTF60W2-II; Haitian, Ningbo, China) to produce plastic bottle caps. The injection molding machine was purchased from Jinfu Technology Co., Ltd. The injection pressure and speed were set to 0.30 s and 80 bar. Temperatures of the injection molding machine were 200 °C, 220 °C, 220 °C, and 220 °C. The schematic flow of the extrusion molding process for the production of the active antioxidant active plastic bottle caps is shown in Figure 1.

### 2.3. Color Attributes

Color changes of the caps with and without antioxidants were measured using a Sphere Spectrophotometer (SP64; X-Rite, Grand Rapids, MI, USA). The machine was calibrated using a standard white tile, according to Equation (1).
(1)$$\mathrm{WI}=100-\sqrt{{\left(100-L\right)}^{2}+a^{2}+b^{2}}$$
where *L*, *a*, and *b* represent lightness, greenness/redness, and blueness/yellowness, respectively, while *WI* was the whiteness index. Each measurement was performed at five different times, with the mean used as the test value.

### 2.4. Thermal Analysis

The effect of antioxidants on the thermal stability of the HDPE matrix was analyzed using differential scanning calorimetry (DSC) (DSC-100; Huanghe, Guangzhou, China). Briefly, 10 mg of the sample was placed on hermetic aluminum pans, under 50 mL·min^−1^ nitrogen atmosphere. The first heating was performed from 25 °C to 250 °C at a rate of 20 °C·min^−1^. The materials were cooled to 25 °C at a rate of 20 °C·min^−1^ before a second heating stage to 250 °C. The degree of crystallinity (*X_c_*), glass transition temperature (*T_g_*), and melting temperature (*T_m_*) were obtained in the second heating stage [12].

### 2.5. Fourier-Transform Infrared (FT-IR) Spectrometry

Fourier-transform infrared (FT-IR) spectra were recorded from 4000 to 400 cm^–1^ at room temperature (FTIR-850; Gangdong, Tianjin, China) to evaluate potential interactions of the antioxidants and HDPE [15].

### 2.6. Torque Tests

The removal and bridge breaking torques were tested using a torque meter (Vortox-i; Mecmesin, Horsham, UK). Briefly, after making the tamper-evident band, the bottle mouths were capped with five sample antioxidant active caps at 1.8 N·m torque using a single head capping machine (provided by Jinfu Technology Co., LTD). Removal torque is the maximum force required to open the tamper-evident plastic cap, whereas the bridge break torque is the maximum force required to break the tamper-evident plastic caps.

### 2.7. Sensory Evaluation

A total of 20 bottles of mineral water (purchased at a local supermarket) had their original caps replaced with antioxidant active caps. Six beakers were filled with Liushen Florida Water (purchased at a local supermarket) and evenly packed inside the carton; then, that mineral water and Liushen Florida Water were stored in the hermetic carton. A panel of six individuals opened the lids at 24 h, 48 h, and 144 h, at which point the water was also poured in a cup and assessed for taste, using normal mineral water as the control. Table 1 categorizes the mean sensory indices.

### 2.8. Antioxidant Assessment

The antioxidant effects of different materials can be measured in several ways. Multiple studies (Table 2) have assessed the antioxidant activities of compounds by measuring the degree of lipid oxidation of oxygen-sensitive foods in packaging materials. The antioxidant activities of plastic bottles are different to those of antioxidant films, which cannot be used for packaging meat and vegetables. Therefore, the antioxidant activity of the caps was tested using three methods: DPPH, ABTS, and FRAP.

The antioxidant effectiveness of the caps was measured according to the 1,1-diphenyl-2-picrylhydrazyl (DPPH) method. DPPH radical-scavenging activity in the presence of the cap extract solution was monitored at 517 nm using a UV/Vis spectrophotometer (Hitachi Ltd., Tokyo, Japan). The film was extracted in methanol at 55 °C for 3 h. The antioxidant activities of the films were determined using the DPPH method. The DPPH scavenging activity was determined according to Equation (2).
(2)$$SA\left(\%\right)=\frac{A_{0}-A_{i}}{A_{0}}\times 100$$
where *A_i_* is the absorbance of the DPPH solution mixed with the cap extract solution, and *A*_0_ is the absorbance of the DPPH solution mixed with methanol.

The ABTS was dissolved in water to reach 7 mM concentration. It was then mixed with 2.45 mM potassium persulfate solution in a 1:1 ratio for 16 h. The solution was then diluted with methanol until it reached the absorbance of 0.70 ± 0.02 at 734 nm. A volume of 40 mL cap extract was added to 3960 mL of the diluted solution. After 6 min incubation in the dark at room temperature, the absorbance was measured at 734 nm.

Free-radical-scavenging activity of the antioxidant caps was determined using the ABTS assay, according to Equation (3).
(3)$$SB\left(\%\right)=\frac{B_{0}-B_{i}}{B_{0}}\times 100,$$
where *B_i_* is the absorbance of a mixture of ABTS and cap extract solution, and *B*_0_ is the absorbance of the ABTS and methanol mixture.

Antioxidant activities of the caps were assessed using the FRAP method. A calibration curve was plotted as a function of the concentration of standard (Vc) solutions (0–1000 ppm). The result was expressed as micrograms of Vc equivalent per gram of film (mg AAE/g film).

### 2.9. Specific Migration Experiment

#### 2.9.1. Migration Conditions

Multiple studies have shown that, under certain conditions, the effect of a short-term but high-temperature process is similar to that of a long-term but low-temperature process. Furthermore, the longer the samples are in contact with solvents, the greater the change is in quality and migration. According to European Union (EU) regulation No.10/2011 [22], antioxidant active plastic bottle caps can be used on juice, water, milk, and alcoholic beverage bottles. The food simulants were as follows: 3% acetic acid, mimicking an acidic food simulant with pH < 4.5; 10% ethanol, mimicking aqueous food simulant; 50% ethanol, mimicking high-alcohol-containing products and milk and some dairy products; 95% ethanol, mimicking a fatty food simulant. Two temperature–time conditions were selected: 40 °C for 10 days and 70 °C for 2 h.

#### 2.9.2. Migration Experiment

The plastic bottle caps were cleaned before the experiment. The irregular shape of the plastic bottle caps could not be tested with one side-contact migration. Therefore, the caps (double-side, 36 cm^2^ in total) were placed into a beaker in which 60 mL of food simulant was added (S/V: 6 dm^2^/1 L). Control experiments without any antioxidants were concurrently performed. All the beakers were incubated under constant temperature and humidity. After the experiment, 2 mL of the food simulant was extracted, filtered using a 0.45 μm organic membrane, and placed in a brown sample bottle for HPLC analysis [23].

#### 2.9.3. Determination and Quantification of Initial Antioxidant Concentrations in Plastic Bottle Caps

The plastic bottle caps were sliced into small pieces (0.3 cm × 0.3 cm, 0.200 ± 0.001 g) and placed in the conical flask. Compounds in the antioxidant active caps were extracted using three different solvents. The conical flask was sealed after adding 20 mL of methanol, methylene chloride, or ethyl acetate for ultrasonic extraction (KQ5200DE; Shumei, Jiangsu, China). The extraction was performed for 3 h at 40 °C. Residues were washed thrice using 20 mL of extraction solvent, and they were later pooled. Thereafter, 10 mL of the supernatant was transferred to a 25 mL test tube and concentrated to 1 mL using sample automatic concentration workstation (Turbo VAP II; Caliper, Princeton, NJ, USA). The water was heated to 40 °C, under an N_2_ pressure of 0.25 MPa. Samples were then diluted with 95% ethanol to the appropriate concentration, filtered using a 0.45 µm organic membrane, and then placed in a brown sample bottle for HPLC analysis. The schematic flow for analyzing the initial contents of the antioxidant active plastic caps is shown in Figure 2.

#### 2.9.4. Chromatographic Analysis of the Antioxidants

We dissolved BHT or BHA in ethanol to prepare the standard solution (1000 µg/mL). The solutions were sequentially diluted using food simulant to 50, 20, 10, 5, 2, 1, 0.5, and 0.2 µg/mL. The solution was stored at 4 °C.

The HPLC analysis was performed using the Agilent apparatus (High-Performance Liquid Chromatography 1260) equipped with a diode array detector (DAD) (G7117C; Agilent, Santa Clara, CA, USA). HPLC was performed in an Agilent SB-C18 column (100 mm × 4.6 mm; i.d.: 2.7 μm). The simulants containing BHA and BHT were analyzed using an isocratic mode of methyl alcohol (*v*/*v*, 85%) and H_2_O–0.1% formic acid (*v*/*v*, 15%) at a flow rate of 0.8 mL·min^−1^, and a column oven (G7130A, Agilent) at 36 ± 0.8 °C. BHT and BHA were detected at 278 and 280 nm, respectively. The autosampler (G7129A, Agilent) injection volume was 4 μL.

#### 2.9.5. Method Validation

Linearity, LOD, LOQ, accuracy, and precision tests were performed to validate the analytical method for the determination of initial concentrations of antioxidants and the migration of BHA and BHT in 3% acetic acid and 10%, 50%, and 95% ethanol. LOD and LOQ were determined as three and 10 times the SD of blank measurements, respectively. Accuracy and precision were evaluated using recoveries and RSDs. Recoveries of the proposed method were verified by spiking BHA and BHT into four food simulants at three concentration levels. The spiked solutions were then treated in the same fashion as samples.

### 2.10. Statistical Analysis

Data were expressed as the mean ± standard deviation (SD). The means were compared using one-way analysis of variance (ANOVA) with a post hoc “LSD test”, using SPSS V. 22 (StatPoint Technologies, Inc., Warrenton, VA, USA) to determine significant differences.

## 3. Results and Discussion

### 3.1. Color Attributes

The cap color determines whether food products can be packed in such materials. The *L**, *a**, *b**, and *WI* properties of the active antioxidant HDPE caps are shown Table 3. Incorporation of BHA and BHT decreased the *L** of the caps by between 1% and 2%, which was statistically insignificant (*p* > 0.05). Given that *L** = 0 represents black while *L** = 100 implies white, the reduction in *L** value implied that the caps became darker. Compared to *L**, the additives were found to significantly affect the *b** and *WI* of composite caps (*p* < 0.05). In particular, the 2% BHA/HDPE and 2% BHT/HDPE compositions significantly induced yellowing of the caps. The antioxidant extract droplets scattered and refracted light, resulting in the darker appearance of the caps [24]. Darker films are ideal for packaging foods sensitive to light [25].

### 3.2. Thermal Analysis

Differential scanning calorimetric analysis for the thermal properties of the antioxidant active caps is shown in Figure 3. Their corresponding melting points, melting enthalpies, and crystallinities are presented in Table 4. As shown in Figure 3, HDPE caps (control) exhibited an endothermic melting of 136.2 °C. Overall, BHA and BHT exhibited an insignificant effect on the *T_m_* and *T_g_* of antioxidant-containing caps. In fact, the antioxidant active caps had a lower *T_m_* compared with HDPE caps, attributed to the plasticizing effect of BHA and BHT (phenolic substance). The antioxidants enhanced the mobility of HDPE chains and consequently the flexibility and ductility of antioxidant active caps [26]. Moreover, BHA and BHT decreased the crystallization of the material sightly, which also influenced the mechanical properties of the antioxidant active caps. Previous studies have documented that active compounds exert a similar effect on packaging materials [27,28].

### 3.3. Fourier-Transform Infrared (FT-IR) Spectrometry

Figure 4a shows the FT-IR spectra of pure antioxidants. Pure BHA and BHT had identical characteristic peaks because of their similar molecular structures. A strong peak at 3628 cm^−1^ (Figure 4a for pure antioxidants) was attributed to –OH vibration in the phenol ring, whereas the absorbance band at 3068 cm^−1^ was attributed to vibrations arising from stretching of C–H bond. Peaks at 2956 and 2872 cm^−1^ represented asymmetric and symmetric stretch vibrations of a methyl group, respectively. Peaks at 815, 765, and 679 cm^−1^ implied the presence of double-bonded carbon atoms (C = C) and in-plane phenol ring bending, while the band observed at 580 cm^−1^ indicated out-of-plane phenol ring bending [29,30].

The FT-IR spectra of the pure HDPE cap and composite BHT and BHA caps at different concentrations are presented in Figure 4b,c. Characteristic peaks of pure HDPE (2917, 2849, 1473, and 719 cm^−1^) were comparable to previous findings [31]. The strength of the bond in the composite caps was evaluated to assess the compatibility between different concentrations of HDPE antioxidants. It was found that the four spectra (i.e., model polymer with or without antioxidants) were almost identical. The physical and chemical interactions of the composites determine the variations of the spectral peaks. The addition of BHT or BHA to the polymer matrix had no effect on the functional groups of the pure HDPE cap, regardless of the concentration. These findings show that the chemical structure of the composite film remained unchanged during granulation, shearing and blowing.

### 3.4. Torque Tests

Plastic antitheft bottle caps are always used on plastic bottled beverage products. Their opening and closing performance torque depends on many factors, such as design, production process, and the capping equipment [32]. The performance of a plastic antitheft bottle cap is generally evaluated on the basis of the removal and bridge break torque [33]. Bottle caps with low bridge strengths have lower opening torque, but can easily come off during transportation. Ideal removal and bridge break torque ranges between 0.8 and 1.4 N·m, which is tight enough to avoid accidental opening but also easy enough for anyone to open the bottle [34,35]. The removal and bridge break torques for the caps tested in this study are presented in Figure 5. BHA and BHT had no significant effect on the two torques (*p* > 0.05), and they were within the qualified range.

### 3.5. Sensory Evaluation

Bottle caps should provide airtight seals. To test this property, the antioxidant plastic bottle caps were used on bottles with pure water placed in a pungent flavor environment. By being airtight, the cap can prevent contamination from gases, water, and other substances. Moreover, the cap should prevent fouling and moisture leakage, as well as maintain a stable internal environment. The performance of the composite caps in comparison to controls is shown in Table 5. Sensory evaluation of the water was performed after days 1, 2, and 6 of storage. There was no significant difference in the taste of water sealed with antioxidant caps (*p* > 0.05). However, all of the mineral water had pungent taste after 144 h of storage. The sensory evaluation system was cheap, easy to perform, and reproducible, characteristics that favor quality food production and management [36]. There was no significant difference in the mineral water taste between the antioxidant active caps and control samples.

### 3.6. Antioxidant Assessment

BHA and BHT have been shown to possess good antioxidant properties. On the other hand, DPPH, ABTS, and FRAP assays are effective in assessing the “total antioxidant capacity” of compounds [37]. For the DPPH assay, a reduction in DPPH due to the effects of BHA and BHT can be characterized by a color change from purple to yellow [38]. Compared to controls, the radical-scavenging capacity of BHA and BHT caps was significantly increased (*p* < 0.05). Specifically, the 2% BHA/HDPE cap exhibited the highest radical-scavenging capacity of 22.02%, which was 10.5 times greater than that of the control cap (Table 6).

The ABTS assay is based on the inhibition of the 2,2-azino-di-[3-ethylbenzthiazoline sulfonate (6)] radical by the antioxidant active caps [39]. Compared to controls, the antioxidant active caps exhibited a significantly high ABTS scavenging activity (*p* < 0.05), regardless of the type (*p* > 0.05). FRAP analysis showed that HDPE caps with either of the antioxidants significantly reduced ferric ion to ferrous ion (*p* < 0.05) even at low concentrations (1% BHA and 1% BHT) (Table 6).

Comparison of the antioxidant effects of the antioxidant active caps showed a significant difference for all the antioxidant assays (*p* < 0.05). We found that the antioxidant activities of BHA and BHT were almost similar, although BHA exhibited a slightly stronger oxidation capacity than BHT, consistent with a previous report [40]. For the antioxidant caps as gauged against the control caps of BHA and BHT, the DPPH method was more suitable for the detection of antioxidant activity.

### 3.7. Specific Migration Experiment

#### 3.7.1. Method Validation

The LODs, LOQs, linear equation, correlation coefficient (*r^2^*), and linear ranges of BHA and BHT antioxidant activities are shown in Table 7. The LODs for both antioxidants ranged from 0.05–0.16 mg/L in 3% acetic acid, 10% ethanol, 50% ethanol, and 95% ethanol. Their correlation coefficient (*r^2^*) was greater than 0.999.

The recoveries and RSD of BHA and BHT at three spiked levels are shown in Table 8. The recoveries of BHA at low (0.4–4 mg/L), intermediate (3–25 mg/L), and high (8–80 mg/L) spiked concentrations ranged from 85.13–106.60%, 93.86–105.42%, and 90.45–108.21%, respectively, whereas those of BHT at low (0.4–4 mg/L), intermediate (4–25 mg/L), and high (8–80 mg/L) concentrations ranged from 88.06–100.22%, 85.21–106.55%, and 84.54–110.92%, respectively. Meanwhile, the RSDs of both BHA and BHT were less than 10%.

#### 3.7.2. Determination and Quantification of Initial Antioxidant Concentrations in Plastic Bottle Caps

Initial antioxidant concentrations in the sample were also assessed (Table 9). It was found that a smaller quantity of the antioxidants was lost during manufacturing of the caps due to thermal effects of extrusion. The theoretical concentrations of antioxidants in active antioxidant caps were 1% (10^4^ mg/kg) and 2% (2 × 10^4^ mg/kg), respectively. This shows that BHT underwent greater thermal decomposition. The extraction capacity of BHA and BHT using the three solvents was also investigated. In the independent experiments, the three methods presented significant differences (*p* < 0.05). The dichloromethane solution exhibited the highest extraction performance, whereas methanol had the worst. Ethyl acetate had a moderate extraction capacity. Methanol has a low polarity that is not suitable for completely extracting BHA and BHT. Methylene chloride and ethyl acetate have good extraction capacities for antioxidants and other phenolic substances. However, methylene chloride extraction is recommended over ethyl acetate.

#### 3.7.3. Migration Experiment and Safety Assessment

Antioxidant activities of BHA and BHT have been extensively studied [41,42]. It has been documented that their antioxidant effects are dependent on the compound structure, concentration of the antioxidant, temperature, light, simulant type, and physical state of the system (e.g., pH). Figure 6 shows the migration of antioxidants from packaging materials to four food simulants at 40 and 70 °C, the migration of BHA and BHT was directly proportional to time and temperature. BHA and BHT did not dissolve in 3% acetic acid or 10% ethanol. These findings suggest that the water solubilities of BHA and BHT are relatively low; thus, they could not migrate in the above solvents [43].

The migration potentials of BHA and BHT in 50% and 95% ethanol at 70 °C are presented in Figure 6a. A 50% ethanol concentration is recommended by the European commission for simulating several dairy products. The migration rates of BHA and BHT in 50% ethanol at 40 °C are presented in Figure 6b. BHA and BHT have equal polarities, but BHA is released faster than BHT. This is because BHT has a higher molecular weight and volume (because of two –C(CH_3_)_3_ groups) [44]. As shown in Figure 6a, the migration potentials of BHA and BHT were the same, even at 40 °C. Overall, the migration of BHT was higher than that of BHA.

Usually, 95% ethanol is used as a simulant for fats, oil, and fatty foods due to their comparable hydrophobicity. Ethanol is completely miscible in water and provides a more hydrophobic environment. Since all the antioxidants used in this study were hydrophobic, they were rapidly released. Additionally, graphs for the rate of antioxidant release with time in comparison to the initial antioxidant amount in HDPE caps were plotted to elucidate on antioxidant release behaviors. The rate of BHA and BHT release in 95% ethanol at 40 °C is shown in Figure 6c. BHA and BHT were found to be rapidly released into simulant foods at the same relative rate. Given the slow dissolution, maximum release was achieved after 24 h. These diverse release behaviors were attributed to the intrinsic properties of the antioxidants, including molecular weight, volume, and polarity, as well as to the extrinsic factors, such as the interaction between the antioxidant and the polymer matrix.

The EU Commission specifies 10% ethanol for simulating alcoholic foods and 3% acetic acid for acidic foods; no migration in these conditions was detected for BHA and BHT, possibly due to the smaller molecular volume of both compounds, which impeded their release into the highly polar simulants. In contrast, more polar food simulants such as water or 3% acetic acid conferred lower release rates. Therefore, the release of antioxidants is often observed in fatty foods such as olive oil, 95% ethanol, or isooctane, unlike aqueous food simulants.

According to (EC) No 1333/2008, BHA and BHT residues in edible oils, fried foods, dried fish products, biscuits, instant noodles, canned fruits, and pickled meat products should not exceed 200 mg/kg. In this study, the specific migration limit (SML) of BHA and BHT at 40 °C for 10 days or 70 °C for 2 h was lower than 0.2 g/kg. Therefore, the antioxidant active plastic bottle caps meet the safely thresholds for food packaging.

## 4. Conclusions

Diffusion of BHA and BHT from HDPE caps to simulants exhibited Fick’s behavior. The higher migration in oil simulant is also a positive feature for the protection of foodstuff with a high fat content, since this kind of food needs better protection from oxidation. In addition, incorporation of BHA and BHT into the antioxidant plastic bottle caps and films had no significant effect on the original excellent HDPE properties. They exhibited good antioxidant properties that could meet the minimum production thresholds. It is very important to control BHA and BHT within a specific migration limit. The antioxidant bottle cap should have good antioxidant properties and ensure the safety of food contact materials. Accordingly, BHT and BHA are promising compounds in the manufacture of active packaging materials. Nonetheless, more studies using typical foods should be performed to validate the suitability of active antioxidant plastic bottle caps in food packaging.

## Figures and Tables

**Figure 1 materials-14-04545-f001:**
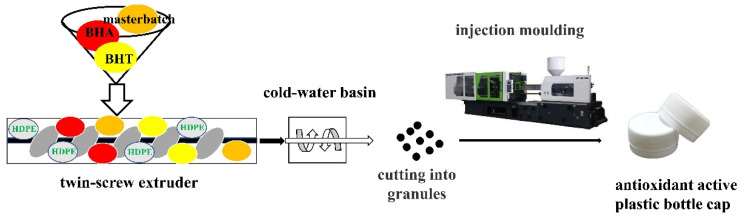
Schematic flow of the extrusion process used in the production of antioxidant active plastic bottle caps.

**Figure 2 materials-14-04545-f002:**
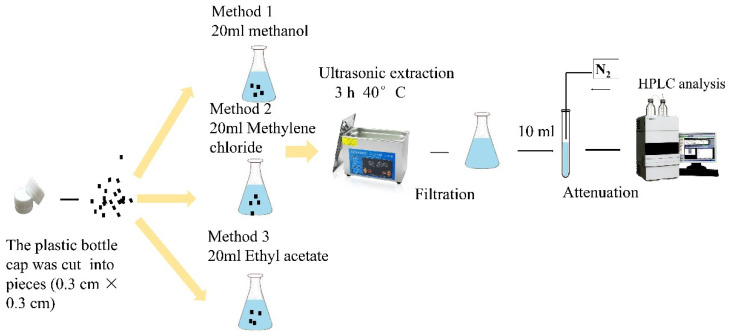
Schematic flow for analyzing the initial contents of the antioxidant in active plastic caps.

**Figure 3 materials-14-04545-f003:**
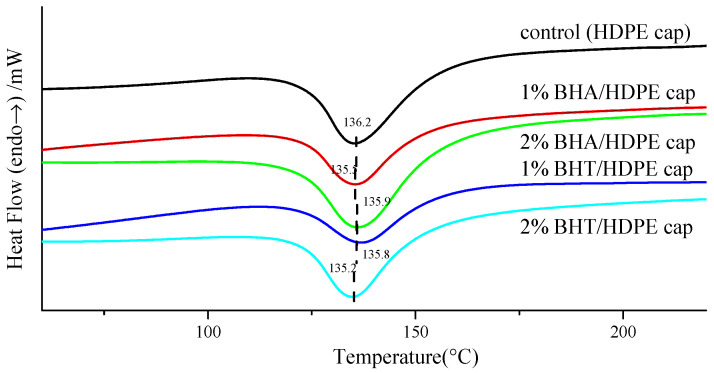
Thermal behavior of antioxidant active plastic bottle caps.

**Figure 4 materials-14-04545-f004:**
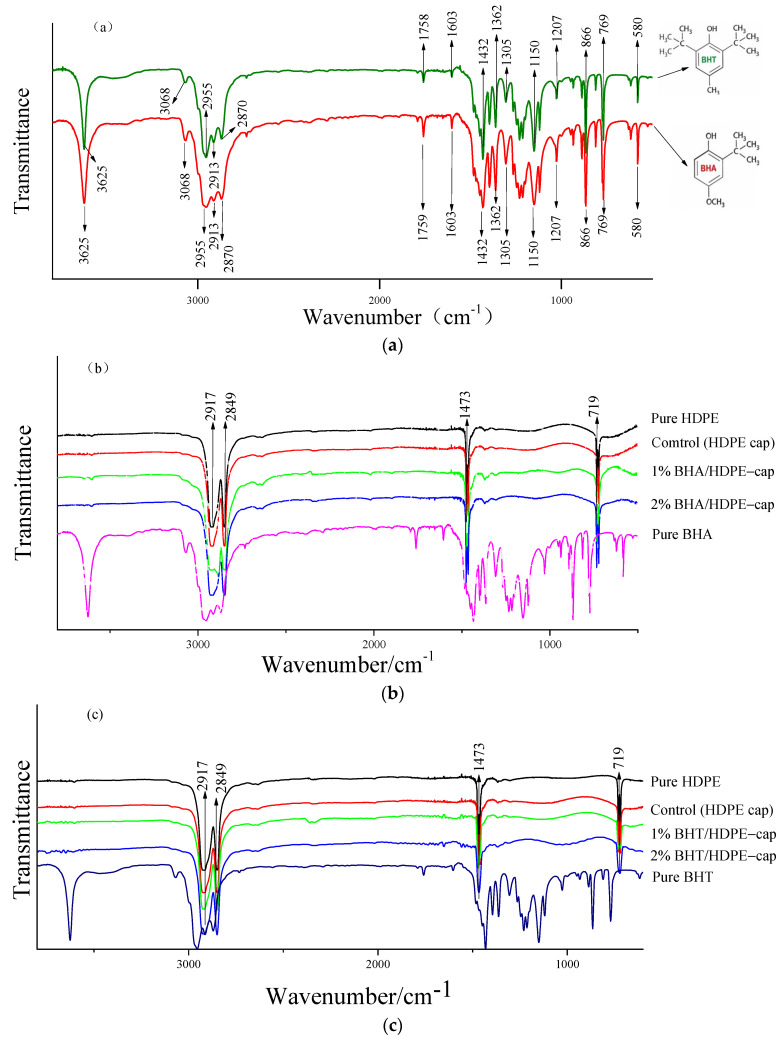
Fourier-transform infrared (FT-IR) spectra of pure antioxidants (**a**) and composite caps containing BHA (**b**) and BHT (**c**).

**Figure 5 materials-14-04545-f005:**
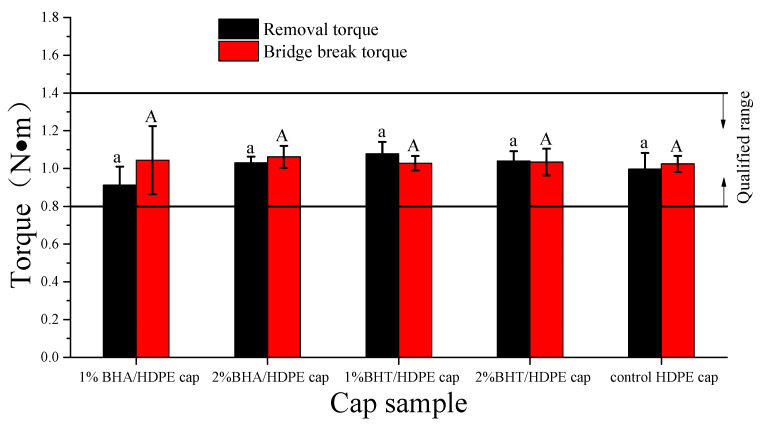
Removal and bridge break torques of antioxidant active plastic bottle caps (*n* = 3).

**Figure 6 materials-14-04545-f006:**
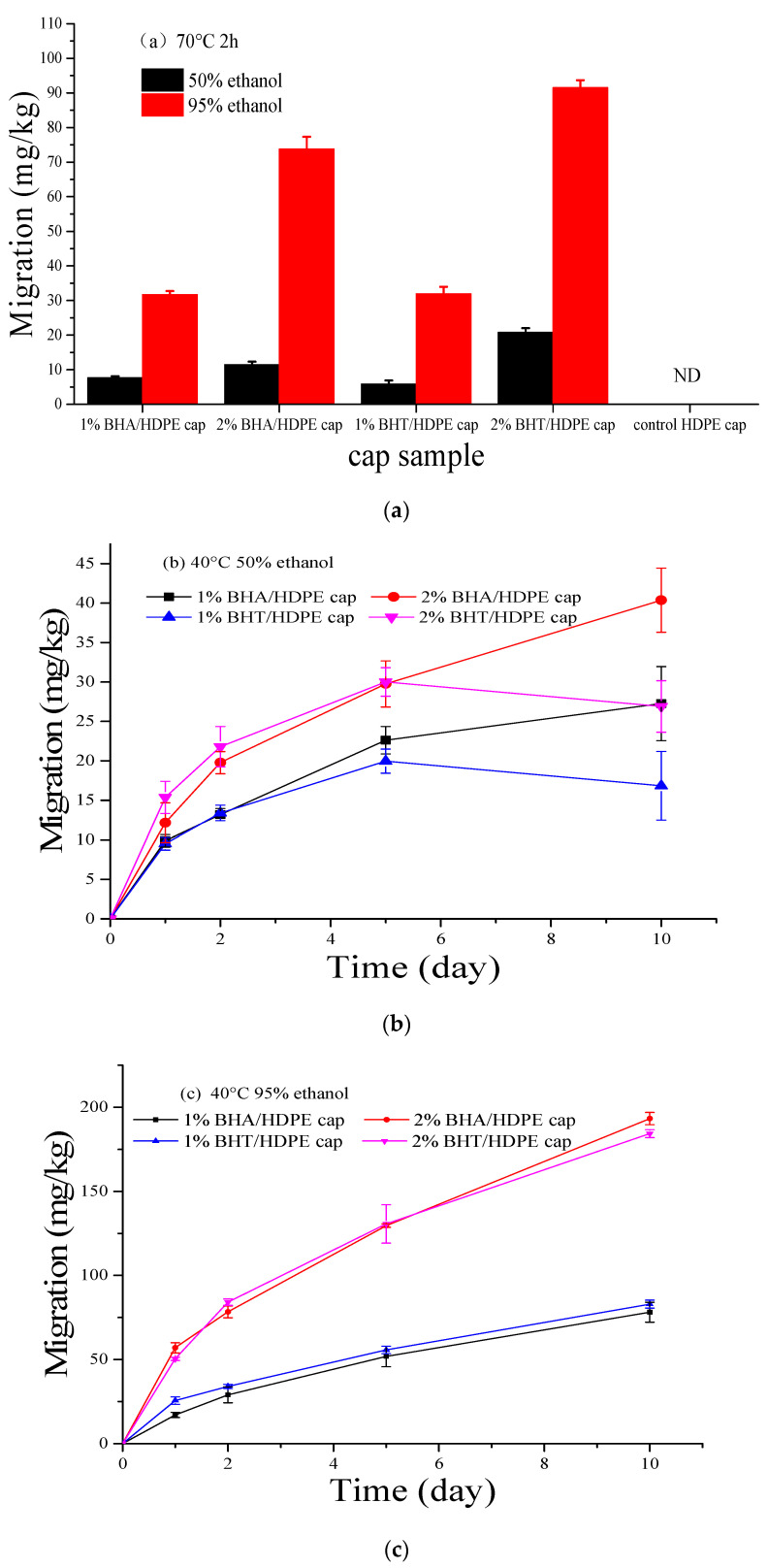
Migration of BHA and BHT from antioxidant active plastic bottle cap into food simulants under the multiple-use condition (*n* = 3), (**a**) in 50% and 95% ethanol at 70 °C, (**b**) in 50% ethanol at 70 °C and (**c**) in 95% ethanol at 70 °C.

**Table 1 materials-14-04545-t001:** The sensory index scoring criteria of antioxidant active plastic bottle caps.

Project	Characteristic	Score
Taste	Taste is normal	9
	The taste is slightly different to that of mineral water	7
	The taste is obviously different to that of mineral water	5
	The mineral water has a heavy taste	3
	The mineral water has a pungent taste	1

**Table 2 materials-14-04545-t002:** Methods for assessing antioxidant activities in active packaging systems.

Application	Method Used	References
Packaging of salmon slices	DPPH, TBARS	[16]
Packaging of fresh meat	MetMb, color, PV, aldehydes	[17]
Nano-biocomposite antioxidant films	DPPH	[18]
Antioxidant chitosan film	DPPH, ABTS, FRAP	[19]
Packaging of frozen fish	TH, CD, TBARS, FRAP, ABTS, color	[20]
Antioxidant polyolefin film	DPPH	[21]

Note: Peroxide values (PV), conjugated dienes (CD), conjugated triene hydroperoxides (TH), thiobarbituric acid index (TBARS), metmyoglobin (MetMb).

**Table 3 materials-14-04545-t003:** The color of antioxidant active plastic bottle caps (mean ± SD, *n* = 5).

Film Sample	*L**	*a**	*b**	*WI*
Control (HDPE cap)	89.57 ± 0.43 ^a^	1.10 ± 0.18 ^a^	3.74 ± 0.51 ^cd^	88.85 ± 0.26 ^a^
1% BHA/ HDPE cap	84.70 ± 1.42 ^b^	1.37 ± 0.40 ^a^	6.01 ± 0.90 ^b^	83.33 ± 0.92 ^c^
2% BHA/ HDPE cap	84.83 ± 1.90 ^b^	1.72 ± 0.02 ^b^	6.79 ± 0.47 ^b^	84.47 ± 1.80 ^bc^
1% BHT/ HDPE cap	83.35 ± 1.09 ^b^	1.41 ± 0.42 ^a^	4.89 ± 0.87 ^bc^	82.59 ± 0.78 ^c^
2% BHT/ HDPE cap	83.20 ± 0.67 ^b^	1.93 ± 0.40 ^b^	7.45 ± 0.23 ^a^	86.72 ± 0.47 ^b^

Note: Different superscript letters in each column denote statistical differences between means (*p* < 0.05).

**Table 4 materials-14-04545-t004:** Thermal parameters of antioxidant active plastic bottle caps based on DSC thermograms.

Cap Sample	*T_g_* (°C)	*T_m_* (°C)	Δ*H_m_* (J/g)	*X_c_* (%)
Control (HDPE cap)	123.93	136.20	149.15	51.91
1% BHA/ HDPE cap	123.40	135.50	145.19	50.54
2% BHA/ HDPE cap	123.40	135.87	142.77	49.69
1% BHT/ HDPE cap	123.93	135.80	146.31	50.93
2% BHT/ HDPE cap	123.90	135.20	142.39	49.56

**Table 5 materials-14-04545-t005:** Sensory index scores for antioxidant active plastic bottle caps (mean ± SD, *n* = 6).

Cap Sample	Taste		
	24 h	48 h	144 h
Control HDPE cap	7.80 ± 1.78 ^a^	6.60 ± 0.89 ^a^	5.40 ± 0.89 ^a^
1% BHA/HDPE cap	7.00 ± 2.00 ^a^	5.80 ± 1.78 ^a^	5.00 ± 0.92 ^a^
2% BHA/HDPE cap	7.40 ± 1.67 ^a^	6.60 ± 1.67 ^a^	5.80 ± 1.09 ^a^
1% BHT/HDPE cap	7.40 ± 1.67 ^a^	6.60 ± 1.67 ^a^	5.80 ± 1.09 ^a^
2% BHT/HDPE cap	7.00 ± 2.00 ^a^	7.00 ± 1.41 ^a^	5.40 ± 0.89 ^a^

Note: Different superscript letters in each column denote statistical differences between means (*p* < 0.05).

**Table 6 materials-14-04545-t006:** Antioxidant effects of the antioxidant active plastic bottle caps (mean ± SD, *n* = 3).

Cap Type	DPPH SA (%)	ABTS SB (%)	FRAP (mg AAE/g DW)
Control HDPE cap	2.09 ± 0.80 ^c^	4.58 ± 1.01 ^c^	21.22 ± 0.79 ^d^
1% BHA/HDPE cap	10.56 ± 1.17 ^b^	27.01 ± 3.61 ^b^	44.13 ± 3.15 ^b^
2% BHA/HDPE cap	22.02 ± 2.31 ^a^	42.10 ± 3.34 ^a^	66.61 ± 1.82 ^c^
1% BHT/HDPE cap	8.26 ± 1.81 ^b^	26.39 ± 1.91 ^b^	43.46 ± 2.78 ^b^
2% BHT/HDPE cap	18.7 ± 1.48 ^a^	43.52 ± 2.01 ^a^	59.33 ± 1.72 ^a^

Note: DPPH radical-scavenging activity (DPPH SA), ABTS radical-scavenging activity (ABTS SB), ferric reducing antioxidant power (FRAP). Different superscript letters in each column denote statistical differences between means (*p* < 0.05).

**Table 7 materials-14-04545-t007:** Linear equations, LOD, and LOQ of BHA and BHT.

Antioxidant	Food Simulant	Linear Equation	Linear Ranges (mg/kg)	Correlation Coefficient (*r^2^*)	LOD (mg/L)	LOQ (mg/L)	Slope Confidence Intervals	Intercept Confidence Intervals
BHA	3% acetic acid	*y* = 1.92*x* − 0.86	0.5–10	0.99908	0.14	0.47	1.88–1.95	−0.89–0.83
	10% ethanol	*y* = 1.15*x* + 0.01	0.5–10	0.99939	0.15	0.47	1.13–1.17	−0.09–0.11
	50% ethanol	*y* = 2.36*x* + 0.24	0.2–50	0.99993	0.06	0.28	2.16–1.56	0.13–0.35
	95% ethanol	*y* = 2.17*x* + 0.52	0.5–100	0.99984	0.16	0.48	2.16–2.18	0.49–0.55
BHT	3% acetic acid	*y* = 1.37*x* + 0.12	0.5–10	0.99948	0.15	0.51	1.34–1.41	0.06–0.18
	10% ethanol	*y* = 1.46*x* + 0.06	0.5–10	0.99957	0.05	0.22	1.44–1.48	0.04–0.08
	50% ethanol	*y* = 2.63*x* + 0.70	0.2–50	0.99992	0.05	0.11	2.61–2.65	0.69–0.71
	95% ethanol	*y* = 2.25*x* − 0.16	0.5–100	0.99991	0.16	0.52	2.18–2.32	−0.52–0.20

**Table 8 materials-14-04545-t008:** The recoveries and RSDs of the BHA and BHT in the migration experiment (mean ± SD, *n* = 3).

Figure	Spiked Concentration (mg/L)		Recoveries (%)	
	BHA	BHT	BHA	BHT
3% acetic acid	0.8	0.8	87.24 ± 3.53	88.06 ± 4.61
	3	6	93.86 ± 1.48	85.21 ± 6.66
	8	8	90.45 ± 1.66	84.54 ± 2.47
10% ethanol	2	1.5	85.13 ± 2.93	93.02 ± 2.94
	6	4	94.65 ± 2.23	94.59 ± 1.49
	8	8	102.13 ± 1.93	99.39 ± 1.25
50% ethanol	0.4	0.4	89.37 ± 0.61	98.76 ± 2.12
	15	15	94.40 ± 2.82	106.55 ± 1.66
	40	40	99.01 ± 4.45	110.92 ± 0.95
95% ethanol	4	4	106.60 ± 1.59	101.22 ± 0.68
	25	25	105.42 ± 0.11	98.89 ± 0.31
	80	80	108.21 ± 0.20	101.35 ± 0.28

**Table 9 materials-14-04545-t009:** BHA or BHT extraction from antioxidant active caps by different solvents (mg/kg, mean ± SD, *n* = 3).

Cap Sample	Extraction Solvent		
	Methanol	Methylene Chloride	Ethyl Acetate
Control HDPE cap	ND	ND	ND
1% BHA/HDPE cap	466.30 ± 34.88 ^a^	3870.55 ± 74.12 ^a^	2961.61 ± 218.92 ^a^
2%BHA/HDPE cap	787.95 ± 50.13 ^b^	6630.82 ± 291.84 ^b^	5991.24 ± 277.72 ^b^
1% BHT/HDPE cap	345.96 ± 11.57 ^c^	2323.18 ± 93.60 ^c^	1940.02 ± 70.59 ^c^
2% BHT/HDPE cap	981.48 ± 45.55 ^d^	3171.66 ± 141.36 ^d^	4283.13 ± 84.64 ^d^

Note: Means in the same column with different superscript letters are significantly different (*p* < 0.05).

## Data Availability

Available upon request from the corresponding author.

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
