# Peer review of "Functional and Antioxidant Properties of Plastic Bottle Caps Incorporated with BHA or BHT"

_materials, 2021, doi:10.3390/ma14164545_

Round 1

Reviewer 1 Report

The manuscript describes the preparation and evaluation of HDPE caps incorporating BHT and BHA. Results focus on the effects on the properties of the material as well as the release of these compounds from the matrix.  In general terms, the article is organized, straight-forward and I cannot find big discrepancies. However, some suggestions will be given in the following lines, to improve the article.

Keywords.

Antioxidant and antioxidant capacity seems quite redundant.

Introduction.

-Lines 44-47. As European legislation is quoted through the test, please include the food groups or categories in which BHT and BHA are authorized.

-Also at lines 44-47. Include references for the antioxidant mechanisms quoted.

-References of studies incorporating these substances in plastics should be provided.

Experimental section.

-Lines 147-149. Please, clarify why oil simulant was not used to test for migration of BHT and BHA, since these substances are hydrophobic and their migration could be enhanced using this food simulant.

-lines 177-178, please, rewrite the sentence: "Standard solutions were prepared by dissolving 0.1000 ± 0.0001 g of BHT or BHA in1000 ug/mL of ethanol."

Results section.

-Please review the information shown in tables 3, 5, 6, 8, and 9. Attention should be paid to the use of significant figures to express both the result and the uncertainty associated with it.

-Please include slope and intercept confidence intervals for linear equation parameters in table 7.

-Line 369: clarify the sentence starting with "At the stated concentrations, the migration of BHA and BHT none detected"

Conclusions

-Authors should discuss whether the migration of BHT and BHA from these caps could be detrimental for those food categories to which these additives are not authorized.  Wouldn't this be a limitation?

Author Response

Dear Editors and Reviewers:

Thank you for your letter and for the reviewers’ comments concerning our manuscript entitled “Functional and antioxidant properties of plastic bottle caps incorporated with BHA or BHT” (ID: materials-1317134).

Those comments are all valuable and very helpful for revising and improving our paper, as well as the important guiding significance to our researches. We have studied comments carefully and have made correction which we hope meet with approval.

Reviewer 1:

  1. Antioxidant and antioxidant capacity seems quite redundant.

Response: We have deleted "Antioxidant Capacity".

  1. Lines 44-47. As European legislation is quoted through the test, please include the food groups or categories in which BHT and BHA are authorized.

Response: We have modified the sentence according to your suggestion.

  1. Also at lines 44-47. Include references for the antioxidant mechanisms quoted. -References of studies incorporating these substances in plastics should be provided.

Response: We have added two references there.

  1. Lines 147-149. Please, clarify why oil simulant was not used to test for migration of BHT and BHA, since these substances are hydrophobic and their migration could be enhanced using this food simulant.

Response: Replacing fatty food simulant with 95% ethanol provides a hydrophobic environment and is cheaper than oil simulant.

  1. lines 177-178, please, rewrite the sentence: "Standard solutions were prepared by dissolving 0.1000 ± 0.0001 g of BHT or BHA in1000 ug/mL of ethanol."

Response: We dissolved BHT or BHA in ethanol to prepare standard solution (1000µg/mL).

  1. Please review the information shown in tables 3, 5, 6, 8, and 9. Attention should be paid to the use of significant figures to express both the result and the uncertainty associated with it. Please include slope and intercept confidence intervals for linear equation parameters in table 7.

Response: We have carefully checked the table and made corresponding modifications. Please refer to the corresponding places in the manuscript for details. 

  1. Line 369: clarify the sentence starting with "At the stated concentrations, the migration of BHA and BHT none detected"

Response: We have modified the sentence to “No migration of BHA and BHT was detected in the 10% ethanol for simulating alcoholic foods and 3% acetic acid for acidic foods”.

  1. Authors should discuss whether the migration of BHT and BHA from these caps could be detrimental for those food categories to which these additives are not authorized. Wouldn't this be a limitation.

Response: BHA and BHT are widely used in the antioxidation of fat foods. Excessive addition of antioxidants will affect health. It is very important to control BHA and BHT in a specific migration limit. The antioxidant bottle cap should have good antioxidant property and ensure the safety of food contact materials. The major challenge associated with adopting food antioxidant packaging materials is an effective and practical combination of the antioxidant and packaging material without changing the packaging line or the characteristics of the materials.

In the introduction and conclusion, we make an additional explanation for it.

Reviewer 2 Report

The paper on application or a trial of application of BHA and BHT as a component of plastic bottle caps is interesting. These antioxidants are frequently applied in processed food  and packing material not only for food but also for cosmetics and personal care products and are approved by EU regulations. The paper is well documented, however, some improvements should be make:

more detailed description of applied methods (DPPH, ABTS and FRAP) for antioxidant activity evaluation should be included 

line 133 - what does it mean: ABTS and methanol mixture I think that description of the method could be helpful

line 136 - as a standard vitamin C was applied, so probably should be mg AAC/g film; GAE is applied for gallic acid equivalents; similar correction is required in Table 6

line 170 - should be MPa

line 171 - should be µm

line 177-178 - should be checked

line 179 - should be µg/mL

line 303-304 in table 6 antioxidant activity for DPPH and ABTS methods are expressed as percent of scavenging activity, and for FRAP as AAE (ascorbic acid equivalent) -it is a different unit, so such sentence: Percentage increase in antioxidant activity... is should be corrected

Figure 6 a, b - ordinate: units should be mg/kg

line 353 - should be subscript

References:

line 410 - should be added number of the paper

line 420-421, 444, 450-451, 453, 459 - abbreviation of journal name should be included/corrected

Author Response

Dear Editors and Reviewers:

Thank you for your letter and for the reviewers’ comments concerning our manuscript entitled “Functional and antioxidant properties of plastic bottle caps incorporated with BHA or BHT” (ID: materials-1317134).

Those comments are all valuable and very helpful for revising and improving our paper, as well as the important guiding significance to our researches. We have studied comments carefully and have made correction which we hope meet with approval.

The main corrections in the paper and the responds to the reviewer's comments are as flowing:

Responds to the reviewer's comments:

Reviewer 2:

  1. more detailed description of applied methods (DPPH, ABTS and FRAP) for antioxidant activity evaluation should be included. what does it mean: ABTS and methanol mixture I think that description of the method could be helpful.

Response: Lines 127-140, We added two methods of experimental operation. Please refer to the corresponding places in the manuscript for details. 

  1. line 136 - as a standard vitamin C was applied, so probably should be mg AAC/g film; GAE is applied for gallic acid equivalents; similar correction is required in Table 6.

Response: Thanks for the reviewer's comments. We have revised it.

  1. line 170 - should be MPa, line 171 - should be µm, line 177-178 - should be checked, line 179 - should be µg/mL.

Response: Thanks for the reviewer's comments. We have revised it.

  1. lines 303-304 in table 6 antioxidant activity for DPPH and ABTS methods are expressed as percent of scavenging activity, and for FRAP as AAE (ascorbic acid equivalent) -it is a different unit, so such sentence: Percentage increase in antioxidant activity. is should be corrected.

Response: For various methods studied in this work,DPPH method is more suitable for the detection of antioxidant activity of antioxidant bottle caps.

  1. Figure 6 a, b - ordinate: units should be mg/kg, line 353 - should be subscript

Response: Thanks for the reviewer's comments. We have revised it.

  1. Response to comment: (References: line 410 - should be added number of the paper, line 420-421, 444, 450-451, 453, 459 - abbreviation of journal name should be included/corrected)

Response: Thanks for the reviewer's comments. We have revised it.